# Patient-Reported Outcomes following Total Knee Replacement in Patients Aged 65 Years and Over—A Systematic Review

**DOI:** 10.3390/jcm12041613

**Published:** 2023-02-17

**Authors:** Nicholas Woodland, Antony Takla, Mahnuma Mahfuz Estee, Angus Franks, Mansi Bhurani, Susan Liew, Flavia M. Cicuttini, Yuanyuan Wang

**Affiliations:** 1School of Public Health and Preventive Medicine, Monash University, Melbourne 3004, Australia; 2Alfred Hospital, Melbourne 3004, Australia

**Keywords:** patient-reported outcomes, total knee replacement, osteoarthritis, pain, function, quality of life, satisfaction

## Abstract

A previous systematic review showed positive patient-reported outcomes following total knee replacement (TKR) in patients aged < 65 years. However, the question remains as to whether these results are replicated for older individuals. This systematic review evaluated the patient-reported outcomes following TKR in individuals aged ≥ 65 years. A systematic search of Ovid MEDLINE, EMBASE, and Cochrane library were performed to identify studies examining disease-specific or health-related quality of life outcomes following TKR. Qualitative evidence synthesis was performed. Eighteen studies with low (*n* = 1), moderate (*n* = 6), or serious (*n* = 11) overall risk of bias were included, with evidence syntheses derived from 20,826 patients. Four studies reported on pain scales, showing improvement of pain from 6 months to 10 years postoperatively. Nine studies examined functional outcomes, showing significant improvements from 6 months to 10 years after TKR. Improvement in health-related quality of life was evident in six studies over 6 months to 2 years. All four studies examining satisfaction reported overall satisfaction with TKR results. TKR results in reduced pain, improved function, and increased quality of life for individuals aged ≥ 65 years. The improvement in patient-reported outcomes needs to be utilised in conjunction with physician expertise to determine what would comprise clinically significant differences.

## 1. Introduction

Osteoarthritis (OA) is a multifactorial joint disease estimated to affect over 300 million people globally, impacting significantly upon patients’ quality of life [1]. The prevalence of OA coincides with increasing age and obesity, with the knee being one of the most frequently affected joints [2]. Conservative management for knee OA is the consensus initial intervention and involves disease education, exercise, weight management, physical therapy, and analgesia [3]. Total knee replacement (TKR) is generally accepted as a cost-effective and worthwhile intervention in the management of end-stage knee OA, utilised and known to improve pain and restore joint function [4,5]. In parallel with the increase in prevalence of OA, the number of TKRs continues to rise [6], with current projections reflecting a continued trajectory and more than 160,000 TKRs anticipated to be performed in Australia in the year 2030 [7].

A prior systematic review and meta-analysis examined the patient-reported outcomes following TKR in patients aged less than 65 years [8]. This report found significant and meaningful improvements in pain, knee function, quality of life, and overall satisfaction with the procedure in individuals within this age demographic [8]. The question remains as to whether these results are replicated for older adults aged over 65 years, a population with increased comorbidities and worse functional and psychosocial status but representing a significant proportion of those undergoing TKR [9,10]. While rehabilitation plays a role in improving pain, function, and mobility outcomes after TKR [11,12], there is evidence that the preoperative conditions of the patients, including pain, functional status, psychosocial factors, and comorbidities, are very important in terms of determining outcomes after TKR [13,14,15,16]. The effectiveness of surgical intervention and patient satisfaction following TKR in patients aged over 65 years remains unclear. However, such evidence is important in order to inform decision-making on the benefits versus risks associated with TKR in this older population who have higher morbidity and mortality [10]. Thus, this systematic review aimed to evaluate the patient-reported outcomes following TKR in patients aged 65 years and over.

## 2. Materials and Methods

This systematic review was conducted in accordance with Preferred Reporting Items for Systematic Reviews and Meta-Analyses (PRISMA) 2020 guidelines [17]. The protocol of the review was prospectively registered with PROSPERO (CRD42021259356).

### 2.1. Search Strategy

MeSH terms and keywords were used to identify studies examining patient-reported outcomes following TKR. The search strategy on Ovid MEDLINE(R) (1946 to December 2022) was developed using Boolean operators “AND” and “OR”. Terms used were “Arthroplasty, Replacement, Knee/”, “Knee Prosthesis/”, “patient outcome assessment/”, “minimal clinically important difference/”, “patient reported outcome measures”, and “treatment outcome”. The strategy on Embase (1947 to December 2022) combined “total knee arthroplasty/or knee replacement/or total arthroplasty/or total knee prosthesis/” with “clinical outcome/or treatment outcome/”, “outcome assessment/or treatment outcome” and “patient-reported outcome/or treatment outcome/”. Cochrane library was also searched using the same question themes to identify additional articles for screening. All searches were limited to English language and full text available only. The references of included articles and published review articles were also searched.

### 2.2. Inclusion and Exclusion Criteria

Study selection implemented the following predetermined inclusion criteria: (1) study participants were adults aged ≥ 65 years; (2) OA was the surgical indication in ≥90% of cases; (3) primary TKR was the surgery undergone; (4) disease-specific or health-related quality of life instrument score as an outcome measure; (5) minimum follow-up of six months. Exclusion criteria were as follows: (1) study participants were aged < 65 years; (2) surgery was a revision of a prior TKR; (3) TKR was undergone for rheumatoid arthritis; (4) non-English language publications; (5) systematic reviews; (6) randomised controlled trials.

### 2.3. Study Selection

Following the search, references were exported into Covidence for screening. Paired review authors (AF, MME, FMC) independently performed title and abstract screening, and then conducted full-text screening. Eligible articles were included for final review following discussion and consensus of the two review authors, with disagreements resolved by YW.

### 2.4. Data Extraction

Two review authors (AT, NW) independently extracted the data using the pre-specified data extraction template, with disagreements resolved by YW. The following data were extracted: (1) study design, (2) patient demographics, (3) surgery characteristics; including percentage of the prosthesis for OA, percentage of the prosthesis with patellar resurfacing, surgery period, and follow-up rate, (4) outcome measures including duration of follow-up, (5) mean/mean difference of the outcomes, (6) minimal clinically important difference, and (7) patient satisfaction. For studies that reported results according to age subgroups, only the results relevant to our age subgroup of ≥65 years were extracted.

### 2.5. Clinically Meaningful Improvements

Where possible, studies were analysed according to the minimal clinically important difference (MCID) of relevant outcome measures. This is a commonly used method to assess whether a change in the scoring system implemented represents a clinically meaningful change for the patient [18]. The MCID represents the difference in scores between patient groups that perceive a minimal but clinically meaningful difference and groups that perceive no difference, with it typically being calculated using either the anchor or distribution methods [19].

### 2.6. Assessment of Methodological Quality

Two authors (AT and YW) independently performed the qualitative risk of bias evaluation using the Risk of Bias in Non-Randomized Studies of Interventions (ROBINS-I) tool [20]. ROBINS-I assesses seven domains: confounding, selection of participants, classification of interventions, deviation from intervention, missing data, outcome measurement, and selective reporting [20]. Disagreements were resolved by consensus from both review authors. There were no disagreements which required involvement of a third review author.

### 2.7. Data Synthesis and Reporting

There was heterogeneity across patient-reported outcomes in terms of the instruments being used, the range of the scoring scale, and the direction of the scale (whether a higher score represents a better or worse outcome). All data were presented according to their original scoring measure, with ranges and explanations tagged alongside the system implemented. Upon consultation with experts, a meta-analysis was deemed to be inappropriate due to the unavailability of the necessary data and variation in the reporting data. Therefore, qualitative synthesis was performed for this systematic review.

## 3. Results

### 3.1. Study Selection

A total of 3975 studies were identified through the search strategy. After removal of duplicates, 3316 articles remained for screening. After title and abstract screening, 285 proceeded to full-text screening, and 267 were excluded with the reasons recorded. Eighteen studies were eligible for inclusion [21,22,23,24,25,26,27,28,29,30,31,32,33,34,35,36,37,38]. No additional articles were found by searching the references of included studies or published review articles. The PRISMA flow diagram illustrates the search results (Figure 1).

### 3.2. Study, Patient and Surgery Characteristics 

Characteristics of the included studies are presented in Table 1. Fourteen studies were prospective [22,25,26,27,28,29,30,31,32,33,34,35,36,37] and four were retrospective [21,23,24,38]. A total of 20,826 patients were analysed through the 18 included studies, with mean age at the time of TKR ranging from 67 to 92 years. The number of study participants ranged from 15 to 8050, with only two studies including less than 100 participants [35,36]. OA was the sole indication for surgery in seven studies [21,26,27,28,31,36,37]. Sixteen studies included a majority of females [21,22,23,24,25,26,27,28,29,30,32,34,35,36,37,38], with only one study having a male majority at 73% [33] and another study not reporting on gender distribution [31]. Five of the studies reported on prosthesis type [25,29,30,34,38], while only three studies reported that the patella was resurfaced [29,30,35]. Patients underwent TKR between the years of 1970 and 2020 with follow-up periods ranging from 6 months to 10 years. Eight studies had a follow-up rate exceeding 90% [21,24,31,32,33,35,36,38].

### 3.3. Assessment of Patient-Reported Outcomes

#### 3.3.1. Patient-Reported Disease-Specific Instruments

All included studies utilised at least one patient-reported disease-specific instrument as presented in Table 1. Knee pain was analysed utilising the Knee Society Pain Score (KSPS) [33,35] and the Oxford Knee Score pain (OKSP) [23,28]. Knee functional outcomes were analysed utilising the Knee Society Function Score (KSFS) [24,30,33,34,35], the Oxford Knee Score function (OKSF) [23,28], the International Knee Score (IKS) function [29], and the HRS (human retirement study) functional status [31]. General knee outcomes were analysed utilising the Knee Society Score (KSS) [24,25,30,33,34,36,38], the Oxford Knee Score (OKS) [22,23,26,27,28], the Western Ontario and McMaster Universities Osteoarthritis Index (WOMAC) [23,25,37,38], the IKS [29], and the Knee Injury and Osteoarthritis Outcome Score (KOOS) [21,37]. Three studies reported on the MCID [21,27,28].

#### 3.3.2. Health-Related Quality of Life Instruments

Six of the included studies reported results on general health-related quality of life (Table 1). Two studies used the SF-12 Physical Component Summary (PCS) and Mental Component Summary (MCS) instrument [28,29], two used the SF-12 PCS alone [25,32], and two used the EuroQol (EQ-5D) instrument [27,37]. Two studies reported on the MCID [27,28].

#### 3.3.3. Patient Satisfaction Instruments

Four studies reported on patient’s satisfaction following TKR (Table 1), using a six-point visual analogue scale (VAS) of eight questions [28], a 100-point VAS for overall satisfaction [27], or a 5-point Likert scale [37]. One study reported on satisfaction but did not specify the instrument used [35].

### 3.4. Assessment of Methodological Quality

A summary of the methodological quality assessment for the included studies is presented in Table 2. Of the 18 included studies, 1 had low overall risk of bias [38], 6 had moderate overall risk of bias [22,23,24,26,29,31], and 11 had serious overall risk of bias [21,25,27,28,30,32,33,34,35,36,37]. The highest risk of bias was seen within the confounding domain, with 11 studies scored as having a serious bias [21,25,27,28,30,32,33,34,35,36,37]. This may be related to a lack of specific analysis performed to control for potential confounding in conjunction with little mention of controlling for potential confounders. All the studies, except one study [38], scored a moderate risk of bias for the outcome measurement domain, due to outcome assessors being aware of the intervention received. A low risk of bias was seen for all the studies for the domains of classification of interventions, deviation from intervention, selective reporting, and selection of participants (except two studies that scored a serious risk of bias [35,37]). The risk of bias across other domains can be attributed to the incomplete outcome data.

### 3.5. Effect of TKR on Pain

Three studies (one moderate risk and two serious risk of bias) examined pain at 6 or 12 months following TKR [23,28,33] (Table 3). The one study with moderate risk of bias reported that OKS pain improved from a preoperative score of 7.8 to a postoperative score of 10.5 in patients aged 70–79 years and from 6.9 to 10.0 in those aged > 79 years at 6 months postoperatively [23]. The two studies with serious risk of bias showed improved OKS pain [28] and KSPS [33] at 12 months after TKR. Two studies, both with serious risk of bias, examined pain at least 2 years after TKR [33,35] (Table 3), showing improvement of KSPS at a minimum of 2 years [35], and at 5 and 10 years [33] postoperatively.

Summary: All three studies reported improved pain at 6 (one study with moderate risk of bias) and 12 months (two studies with serious risk of bias) following TKR. Two studies (serious risk of bias) reported improved pain at 2–10 years following TKR.

### 3.6. Effect of TKR on Functional Outcomes

Seven studies examined the functional outcomes at 6 or 12 months following TKR [23,24,28,29,30,33,34] (Table 3). Four studies (one moderate risk and three serious risk of bias) examined KSFS [24,30,33,34]. The one study with a moderate risk of bias showed an increase in KSFS from 45 to 55 at one year after TKR [24]. The three studies with serious risk of bias also reported improved KSFS at one year postoperatively [33,34], and higher improvement in the bilateral TKR subgroup compared with the unilateral TKR subgroup at six months and one year postoperatively [30]. Two studies used OKS function (one moderate risk and one serious risk of bias). The study with moderate risk of bias reported an improvement at six months postoperatively, from 12.5 to 16.7 in patients aged 70–79 years and from 11.6 to 15.4 in those aged > 79 years [23]. Another study with serious risk of bias reported improved OKS function in patients aged ≥ 80 years and those aged 65–74 years at 12 months postoperatively [28]. In another study with moderate risk of bias, significant improvement in IKS function score was reported with a median improvement of 25 (IQR 10–35) at 12 months postoperatively [29].

Five studies examined the functional outcomes at least two years after TKR [24,31,33,34,35] (Table 3). Four studies examined KSFS (one moderate and three serious risk of bias). The study with moderate risk of bias reported improvement of median KSFS from 45 to 55 at 3 years, but with no difference in the preoperative score at 5 and 10 years after TKR [24]. Two studies with serious risk of bias showed significant improvement of KSFS at 5 [33,34] and 10 years [33] postoperatively, while another study with serious risk of bias showed non-significant KSFS improvement over 3.9 years follow-up [35]. Using a different tool, another study with moderate risk of bias showed an improved functional outcome at 1.51 years after TKR, in terms of mobility (0.130, SD 1.694), gross motor function (0.012, SD 1.255), and activities of daily living (−0.084, SD 0.908) [31].

Summary: All seven studies reported improvement of functional outcomes following TKR at 6 (one study with moderate and one study with serious risk of bias) and 12 months (two studies with moderate and four studies with serious risk of bias). There was inconclusive data among the five studies examining functional outcomes at 2–10 years following TKR (two studies with moderate and three studies with serious risk of bias), all reporting improvement, except for one study (moderate risk of bias) showing improvement at 3 years but no improvement at 5 and 10 years [24].

### 3.7. Effect of TKR on General Knee Outcomes

Eleven studies examined the general knee outcomes at 6 or 12 months following TKR [22,23,24,27,28,29,30,33,34,37,38], using KSS [24,30,33,34,37,38], OKS [22,23,27,28], WOMAC [23,38], IKS [29], and KOOS [37] (Table 4). Six studies examined KSS (one low risk, one moderate risk, and four serious risk of bias). The only study with low risk of bias reported significant KSS improvement from 48 to 92 at 12 months after TKR, with greater improvement in the octogenarian group compared with the non-octogenarian group [38]. The study with moderate risk of bias reported that KSS improved from 29 to 91 at one year after TKR [24]. The four studies with serious risk of bias also showed improved KSS at one year postoperatively [33,34,37], and greater improvement in the bilateral subgroup compared with the unilateral subgroup at 6 months and 12 months post operatively [30]. Four studies examined OKS [22,23,27,28]. Three studies (one moderate and two serious risk of bias) showed improved OKS at 6 months postoperatively. The study with moderate risk of bias showed an improvement from 20.3 to 27.3 in patients aged 70–79 years and from 18.2 to 26.5 in those aged > 79 years [23]. The two studies with serious risk of bias reported a linear trend for greater improvement with younger age [27], or similar improvement in patients aged ≥ 80 years and those aged 65–74 years [28]. Two studies (one moderate [22] and one serious [28] risk of bias) showed improved OKS at 12 months postoperatively, with an increase from 17.5 to 34.5 in patients aged 70–80 years and from 17 to 32.5 in those aged > 80 years [22]. A study with low risk of bias reported improvement of WOMAC from 48 to 78 at 12 months after TKR [38], and another study with moderate risk of bias showed improved WOMAC from 46.6 to 63.5 in patients aged 70–79 years and from 49.2 to 63.5 in those aged > 79 years at six months postoperatively [23]. Another study with moderate risk of bias reported a median IKS improvement of 71 (IQR 47–92) from a median score of 72 to 142 at 12 months after TKR [29]. One study with serious risk of bias showed significantly improved KOOS scores at 12 months after TKR [37].

Nine studies examined the general knee outcomes at two or more years after TKR [21,22,24,25,26,27,33,34,36], using KSS [24,25,33,34,36], OKS [22,26,27], KOOS [21], and WOMAC [25] (Table 4). Five studies (one moderate and four serious risk of bias) examined KSS. The study with moderate risk of bias reported an improvement from 29 to 92 at 3 years, 91 at 5 years, and 90 at 10 years after TKR [24]. The four studies with serious risk of bias showed improved KSS at more than 2 years [25], 4.6 years [36], 5 years [33,34], and 10 years [33] postoperatively. All three studies examining OKS (two moderate and one serious risk of bias) showed improvement of OKS. One study with moderate risk of bias showed an increase in OKS from 17.5 to 32.5 at 5 years and 29.5 at 10 years postoperatively in patients aged 70–80 years, and from 17 to 31 at 5 years and 28.5 at 10 years postoperatively in those aged > 80 years [22]. Another study with moderate risk of bias reported yearly OKS until nine years postoperatively in patients aged 70–80 years and >80 years, which was significantly higher than the preoperative score at all time points [26]. The study with serious risk of bias showed a linear trend for greater improvement with younger age at two years after TKR [27]. Another two studies with serious risk of bias reported improvement in all the KOOS domains at two years after TKR [21] and improved WOMAC at follow-up more than two years after TKR [25].

Summary: All eleven studies reporting on general knee outcomes showed improvements of postoperative scores at 6 (one study with moderate and three studies with serious risk of bias) and 12 months (one study with low risk of bias, three studies with moderate and five studies with serious risk of bias) from baseline OKS, KSS, WOMAC, IKS, and KOOS measures. All nine studies (three with moderate and six with serious risk of bias) reporting on general knee outcomes showed improvements in 2–10 years postoperative scores from baseline OKS, KSS, KOOS, and WOMAC measures.

### 3.8. Effect of TKR on Health-Related Quality of Life

Five studies examined health-related quality of life at 6 or 12 months following TKR [27,28,29,32,37] (Table 5). Three studies (one moderate risk [29] and two serious risk of bias [28,32]) examined SF-12 PCS, all reporting improvement of SF-12 PCS scores at 12 months postoperatively. The study with moderate risk of bias showed an improvement of 9.47 (IQR 3.24–18.84) from 25.41 to 35.84 [29]. Two studies examined SF-12 MCS following TKR, showing no significant changes at 12 months [28,29]. Two studies with serious risk of bias examined EuroQol [27,37], reporting significant improvement in EQ-5D and EuroQoL VAS at 6 [27] and 12 months [37] after TKR, with a linear trend for greater EQ-5D improvement with younger age at 6 months [27].

Two studies with serious risk of bias examined health-related quality of life at least two years after TKR [25,27] (Table 5). One study reported improvement of EQ-5D and EuroQoL VAS at two years after TKR in all age groups, and a linear trend for greater improvement in EQ-5D with younger age [27]. Another study reported an improvement of SF-12 PCS at least 2 years after TKR [25].

Summary: All five studies reported improvement of health-related quality of life at 6 (one study with serious risk of bias) and 12 months (one study with moderate and three studies with serious risk of bias) following TKR, whereas SF-12 MCS did not improve significantly. Two studies (serious risk of bias) reporting on health-related quality of life at two years following TKR exhibited improvement in outcome measures.

### 3.9. Clinically Meaningful Outcomes following TKR

Clinically meaningful outcomes of TKR were reported in three studies (all with serious risk of bias) using distribution-based [21,27,28] or anchor [21,27] methods. Two studies reported improvement in OKS greater than the MCID [27,28], with the proportion of patients achieving a clinically meaningful improvement at 6 months and 2 years greater in the younger groups [27]. Another study reported on specific percentage results with 77–91% achieving MCID using the distribution-based methods and 76–81% achieving MCID using the anchor-based method for KOOS domains [21]. In terms of quality of life, one study reported improvement in SF-12 PCS greater than the MCID, but the mean difference between the groups (≥80 years vs. 65–74 years) did not exceed the MCID [28]. Another study reported improvement in EQ-5D greater than the MCID at 6 months and 2 years after TKR, with younger patients more likely to achieve a clinically meaningful improvement at 2 years [27].

Summary: All three studies (serious risk of bias) which utilised a MCID reported clinically meaningful improvement following TKR.

### 3.10. Satisfaction with the Results of TKR

Four studies (all with serious risk of bias) examined satisfaction using different tools [27,28,35,37] (Table 6). In Wendelspiess’s study, 89% of patients reported satisfaction [37]. In the study by Clement et al., the overall satisfaction score was 17.4, with 19.0 (SD 8.1) in patients aged > 80 years and 18.2 (SD 8.1) in patients aged 65–74 years, where the total score ranged from eight (most satisfied) to 48 (least satisfied) [28]. In Williams’s study, the overall mean satisfaction score ranged from 83.6 to 85.2 for surgical outcome for patients aged 65–74 years, 75–84 years, and ≥85 years, where a score of 100 indicated most satisfied [27]. Pagnano et al. reported that all but one patient was satisfied with the result of the TKR [35].

Summary: The four studies (serious risk of bias) which reported on patient satisfaction with their TKR provided positive results.

## 4. Discussion

The findings in our systematic review support TKRs as a viable choice for the management of primary OA for elderly individuals aged ≥ 65 years. The outcomes measured by disease-specific and general health instruments consistently demonstrated significant improvements in pain, joint function, and health-related quality of life following TKR. Patient satisfaction levels were also high following TKR, based on the results from the four studies which reported on these measures [27,28,35,37]. The overall findings suggest that TKRs are an appropriate intervention for the management of knee OA in older adults aged ≥ 65 years.

TKR showed improvements in all four studies (one with moderate [23] and three with serious risk of bias [28,33,35]) reporting on pain scales over a follow-up period from 6 months up to 10 years postoperatively, with improvements across all the time points [23,28,33,35]. Of the nine studies (four with moderate [23,24,29,31] and five with serious [28,30,33,34,35] risk of bias) examining functional outcomes using different instruments, large improvements in functional outcomes were observed with follow-up durations ranging from 6 months to 10 years [23,24,28,29,30,31,33,34,35]. However, one study reported a reduction in function from 5 to 10 years after TKR using the KSFS [24], suggesting some uncertainty and the need for further work to clarify this. Uncertainty also remains for the benefits of TKR in improving pain and joint function beyond the first decade following intervention. Such information is needed to guide decision making and generate realistic expectations of the potential results following TKR.

Improvements in health-related quality of life were evident in six studies (one with moderate [29] and five with serious [25,27,28,32,37] risk of bias) over a follow-up period ranging from six months to two years, assessed using EQ-5D and SF-12 PCS scores. However, SF-12 MCS scores did not improve significantly [28,29], suggesting TKR might predominantly improve physical health rather than mental health. Three studies (all with serious risk of bias) reported patient-reported outcomes in relation to MCID of OKS [27], KOOS [21], SF-12 PCS [28], and EQ-5D [27], showing clinically meaningful outcomes of TKR. One study showed that younger patients (aged 65–74 years) were more likely to achieve clinically important improvement in OKS and EQ-5D compared with older patients (aged ≥ 75 years) [27], while another study found no clinically important difference in the improvement of OKS and SF-12 PCS between patients aged ≥ 80 years and those aged 65–74 years [28]. The effect of age on clinically important improvement of patient-reported outcomes following TKR warrants further investigation.

All four studies (with serious risk of bias) examining satisfaction exhibited overall patient-reported satisfaction with TKR procedure [27,28,35,37]. These satisfaction results are consistent with the broader literature and studies analysing TKRs for younger patients [8,44,45]. Despite these findings, the measure of satisfaction is under the influence of a range of factors and thus remains a complex area. The fact that up to 25% of patients reported dissatisfaction following TKR [46] highlights the need for healthcare providers to establish clear patient expectations for the intervention and acknowledge the longer-term uncertainties that remain due to the lack of clinically significant long-term follow-up measures.

TKRs have been shown to be a successful intervention for the management of end-stage OA and their implementation is now global. In patients aged ≥ 65 years, decisions to undergo TKR need to be scrutinised against the risk of the surgery and the influence of any comorbidities upon the procedure, implementation of anaesthesia and inevitable rehabilitation period, including the increased risks with every decade increase in age. Preoperative physical function, mental health, and comorbidities, as well as blood loss also affect the results of TKR [13,14,15,16,47,48,49,50]. Overall the results of this systematic review support the effectiveness of TKR for improving pain, function, and quality of life. One study, albeit of serious risk of bias, showed greater improvements in general knee outcomes and health-related quality of life in patients aged < 75 years compared with those aged ≥ 75 years and that the younger patients were more likely to achieve clinically meaningful improvements [27]. This is especially important as higher morbidity and mortality are seen in elderly patients following TKR compared to younger patients [10]. There is a reduced life expectancy following replacement in this demographic in comparison to a younger patient demographic. Thus, the chances of revision surgery due to prosthesis depreciation are less likely and may not be as large of an influencing factor in whether the older patient proceeds with surgery. An interesting trend identified was the increased functional impairment from 5 to 10 years in patients who were followed up to the 10 years postoperative mark [24]. Importantly, this measure was still greater than baseline, but may suggest that with advancing age, function will taper. This phenomenon can be speculated to be a result of an increasingly sedentary lifestyle and the increase in comorbidity burden and not necessarily as a direct result of the TKR. It may also highlight the need for ongoing rehabilitation to optimize long-term outcomes. Uncertainty surrounding longer-term outcomes and the potential pattern of functional decline should be further examined in future studies.

There are strengths and limitations in our systematic review. This study is the first systematic review examining patient-reported outcomes following TKR for primary OA in patients aged ≥ 65 years. This demographic is set to be ever growing for TKR recipients as waiting lists for elective surgical procedures are at a near all-time high following the impact of the COVID-19 pandemic. The findings from this review may assist clinicians and patients by further clarifying expectations of outcomes following TKR. Only 1 of the 18 included studies had low risk of bias; 6 studies had moderate and 11 had serious risk of bias. Nevertheless, there was general consistency of findings. Although the data showed consistent improvement in outcomes using disease-specific and generic health instruments, the variety in such instruments and the large degree of heterogeneity amongst the included studies deemed meta-analysis of results inappropriate. The limitation of these scores is that no consensus exists as to what improvement in score is clinically meaningful for the patients [21]. The clinical significance of these results should be further investigated and clarity sought on how much patient or procedural characteristics contribute to the improvements in the patient-reported outcomes. The results from the included studies are from a vast array of settings and thus direct comparison should be made with caution and unique circumstances within individual studies scrutinized. In addition, we did not examine the effect of rehabilitation on patient outcomes after TKR in this study. Rehabilitation is recommended for TKR [51] but is a very complex area with recent reviews examining different timing, type, and duration [11,12,52]. As these studies have been on patients with primary OA, the results cannot be extrapolated to other diseases of the knee such as inflammatory or injury-induced arthritis. There is a lack of evidence beyond the first decade of follow-up following TKR. Additional research into patient-reported outcomes following the first decade post intervention as well as further studies utilising disease-specific instruments would assist in gaining knowledge regarding the clinical significance of improvements following TKR and thus guide future management.

## 5. Conclusions

For elderly individuals aged 65 years and over with primary OA, TKR results in reduced pain, improved function, and increased quality of life. The improvement in patient-reported outcomes needs to be utilised in conjunction with physician expertise to determine what would comprise clinically significant differences. It is important to establish the expected outcomes of TKRs in the ever-increasing elder population to assist in guiding healthcare decisions for doctors and patients, and the awareness that patients may face significant periods of rehabilitation in order to optimize patient outcomes. This will allow for evidence-based decisions and aid in determining whether the intervention is likely to yield outcomes which would be of benefit to the patients. Individual patient’s risk of surgical adverse reactions must also be considered.

## Figures and Tables

**Figure 1 jcm-12-01613-f001:**
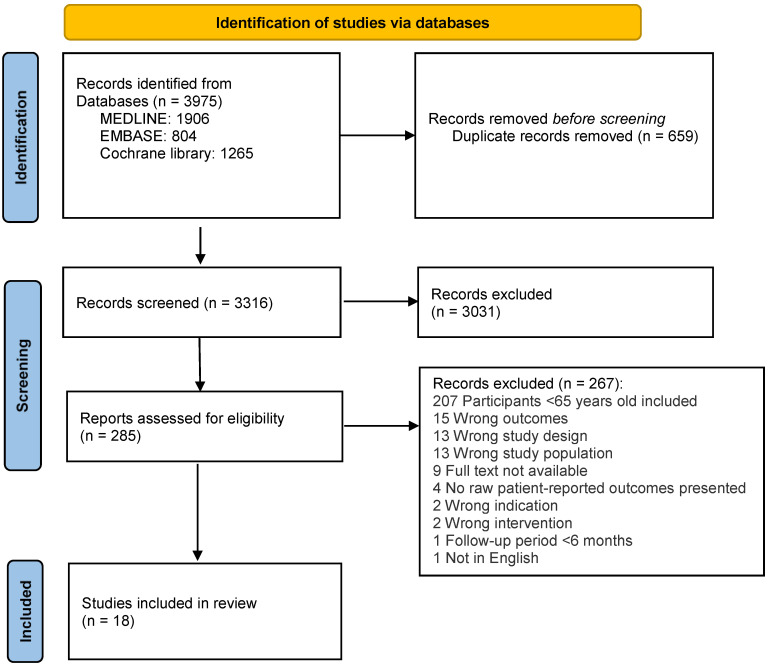
PRISMA flow diagram of included studies.

**Table 1 jcm-12-01613-t001:** Characteristics of included studies.

Study (Author, Year, Country)	Study Design	No of Patients (TKRs)	% for OA	Prosthesis, Patellar Resurfacing	Age (Mean/Median/Range) % of Female	Surgery Period	Follow-up Duration (Range)	Follow-Up Rate	Disease Specific Instruments	Generic Health Instruments	Satisfaction Instruments
Leung et al., 2022. China [38]	Retrospective case–control study	112	99.1	posterior-stabilised (PS) 100%, N/A	Mean 82.7 (SD 1.6) 71.4% female	2011–2015	12 months	100%	Knee Society Score (KSS), Western Ontario and McMaster Universities Osteoarthritis Index (WOMAC)	N/A	N/A
Wendelspiess et al., 2022. Switzerland [37]	Prospective cohort study	218 patients (243 TKRs) 166 patients aged ≥ 65 years	100	N/A, N/A	Customised individually made TKR group: Mean 67.1 (SD 8.4), 45% female Off-the-shelf TKR group: Mean 69.7 (SD 8.9), 60% female	2017– 2020	12 months	79%	KSS, Knee Injury and Osteoarthritis Outcome Score (KOOS)	EuroQol (EQ-5D-3L)	5-point Likert scale
Lyman et al., 2018. United States [21]	Retrospective study	2630 patients	100	N/A, N/A	Mean 74 (SD 6) 63% female	2007–2012	2 years	100%	KOOS	N/A	N/A
Townsend et al., 2018. United States [23]	Retrospective chart review	118 patients	95	N/A, N/A	70–79 group: 62% female >79 group: 54% female	2008-2014	180 days	N/A	Oxford Knee Score (OKS), OKS pain (OKSP), OKS function (OKSF), WOMAC	N/A	N/A
Jiang et al., 2017. United Kingdom [22]	Prospective cohort study	2080 patients (total population, no numbers for those aged > 70 years)	95.2	N/A, N/A	Mean 70 (SD 8) 56.4% female (total population, no numbers for those aged > 70 years)	1999–2003	1 year 5 years 10 years	1 year: 75.8% 5 years: Not reported 10 years: 41.6%	OKS	N/A	N/A
Kennedy et al., 2013. United Kingdom [24]	Retrospective cohort study	438 patients	95.7	N/A, N/A	>80 (mean 83) 54.6% female	1995–2005	6 (5–15.5) years	100% at 5 years after surgery 82% at a mean of 7 years after surgery	KSS, Knee society function score (KSFS)	N/A	N/A
McCalden et al., 2013. Canada [25]	Prospective cohort study	2709 patients	97.3	82% PS & 18% cruciate-retaining (CR), N/A	>70 (mean 77, SD 5) 61% female	1996–2009	Minimum 2 years	N/A	WOMAC, KSS	SF-12 PCS	N/A
Williams et al., 2013. United Kingdom [27]	Prospective cohort study	65–74 years: 865 patients 75–84: 760 >85: 124 Total 1749 patients	100	N/A, N/A	65–74 group 60.8% female 75–84 63.5% female >85 65.4% female	2006–2008	6 months 24 months	6 mths 65–74: 88.8% 75–84: 88.0% >85: 89.2% 24 mths 65–74: 41.4% 75–84: 43.6% >85: 40.0% (very small % of UKR included)	OKS	EQ-5D	100-point visual analogue scale
William et al., 2013. United Kingdom [26]	Prospective cohort study	1266 patients (1547 TKRs)	100	N/A, N/A	Mean 71.5 (SD 8.7) 61.4% female	1994–2008	10 years	N/A	OKS	N/A	N/A
Clement et al., 2011. United Kingdom [28]	Prospective cohort study	Case 163 patients (185 TKRs) Control 435 patients (492 TKRs)	100	N/A, N/A	Case group (>80 years): mean 83.3, range 80–92 63.2% female Control group (65–74 years): mean 70.7, range 65–74 55.1% female	2006–2008	6 months 12 months	N/A	OKS	SF-12 (PCS and MCS)	8 questions with each question having a graded response using a visual analogue scale (1–6)
Dowsey et al., 2010. Australia [29]	Prospective cohort study	211 patients	91.5	CR 88, PS 119, ultra-congruent 4, 37% had patellar resurfacing	Median 74, IQR 68 to 79 60% female	2006– 2007	12 months	N/A	IKS (International knee score)	SF-12 (PCS and MCS)	N/A
Severson et al., 2009. United States [30]	Prospective cohort study	312 patients with unilateral TKR and 70 patients with bilateral TKR	98	PS 100%, 312 (100%) had patellar resurfacing	Unilateral group: mean 76.9 years (SD 4.89), 74% female Bilateral group: mean 77.2 years (SD 5.12), 73% female	N/A	6 months, 1 year	N/A	KSS, KSFS	N/A	N/A
Sloan et al., 2009. United States [31]	Prospective cohort study	516 patients	100	N/A, N/A	>65 (no specific mean or range reported) % female not reported	1994–2006	1.51 years (maximum 4 years)	100%	HRS functional status measures (based on physical function measures from the SF-36, and include additional measures for ADL limitations): Mobility, Gross motor skills, Large muscle, Activities of daily living	N/A	N/A
Franklin et al., 2008. United States [32]	Prospective cohort study	8050 patients (total population, no numbers for those aged >65 years)	95	N/A, N/A	Mean 68 (SD 10) 66% female (total population, no numbers for those aged > 65 years)	2000–2005	12 months	100%	KSS (results not reported for patients aged > 65 years)	SF-12 (PCS)	N/A
Whiteside and Vigano. 2007. Italy [33]	Prospective cohort study	122 patients (167 TKRs)	97	N/A, N/A	Mean 72, range 67–83 27% female	1993–2000	7.3 (5–10) years 1year 5 years 10 years	98%	KSS, Knee Society pain score (KSPS), KSFS	N/A	N/A
Dixon et al., 2004. Australia [34]	Prospective cohort study	135 patients	91	CR 100%, N/A	Mean 79, range 75–83 59% female	1992–1997	7.1 (5–10) years 1 year 5 years	N/A	KSS	N/A	N/A
Pagnano et al., 2004. United States [35]	Prospective cohort study	34 patients (44 TKRs)	91	N/A, 44	Mean 92, range 90–102 67.6% female	1970–1997	3.9 years (minimum 2 years)	97%	KSPS, KSFS, Functional results (instrument not specified)	N/A	Satisfaction assessed but instrument not specified
Tankersley and Hungerford. 1995. United States [36]	Prospective cohort study	15 patients (20 TKRs)	100	N/A, N/A	Mean 87, range 86–95 86.7% female	1980–1991	4.6 years (3–12 years)	100%	Modified KSS	N/A	N/A

N/A: Not applicable.

**Table 2 jcm-12-01613-t002:** Assessment of methodological quality.

Study	Confounding	Selection of Participants	Classification of Interventions	Deviation from Intervention	Missing Data	Outcome Measurement	Selective Reporting	Overall Risk of Bias
Leung et al., 2022 [38]								
Wendelspiess et al., 2022 [37]								
Lyman et al., 2018 [21]								
Townsend et al., 2018 [23]								
Jiang et al., 2017 [22]								
Kennedy et al., 2013 [24]								
McCalden et al., 2013 [25]					N/A			
Williams et al., 2013 [27]								
Williams et al., 2013 [26]								
Clement et al., 2011 [28]					N/A			
Dowsey et al., 2010 [29]								
Severson et al., 2009 [30]								
Sloan et al., 2009 [31]								
Franklin et al., 2008 [32]								
Whiteside and Vigano. 2007 [33]								
Dixon et al., 2004 [34]					N/A			
Pagnano et al., 2004 [35]								
Tankersley and Hungerford. 1995 [36]								

Red: serious risk of bias; orange: moderate risk of bias; green: low risk of bias.

**Table 3 jcm-12-01613-t003:** Effect of total knee replacement on pain and functional outcomes.

Study	Pre-Operative		Post-Operative		Difference		MCID			Summary of Statistical Significance	Summary of Clinical Significance
	Mean	SD	Mean	SD	Mean	SD	MCID (95% CI)	Method	Reference		
**PAIN**											
**Oxford Knee Score (OKS) pain (OKSP) (0–20) *0 being worst, 20 being best***											
Townsend et al. [23]	70–79 years: 7.8 >79 years: 6.9	N/A	70–79 years: 10.5 >79 years: 10.0	N/A	70–79 years: 2.7 >79 years: 3.1	N/A	N/A	N/A	N/A	N/A	N/A
Clement et al. [28]	Not reported	N/A	Not reported	N/A	Case (≥80 years): 7.3 Control (65–74 years): 7.7	Case: 4.6 Control: 4.6	3	Distribution-based	Clement et al., 2011 [28]	In both groups, the postoperative OKSP scores at 12 months were significantly improved (*p* < 0.05). Improvement in OKSP scores showed no significant difference between the groups.	The improvement in OKSP scores was significant relative to preoperative scores and was greater than the MCID. The 95% CI for the mean difference in improvement between the groups did not differ by more than the MCID, suggesting there is no clinical difference in outcomes between groups.
**Knee society pain score (KSPS) (0–100) *0 being worst, 100 being best***											
Whiteside and Vigano [33]	30	6	1 year: 43 5 years: 47 10 years: 48	1 year: 8 5 years: 4 10 years: 3	N/A	N/A	N/A	N/A	N/A	KSPS significantly improved at 1-, 5-, and 10-year follow-ups.	N/A
Pagnano et al. [35]	30	Range 2–66	86	Range 32–99	N/A	N/A	N/A	N/A	N/A	KSPS improved significantly (*p* < 0.01).	N/A
**FUNCTIONAL OUTCOMES**											
**Knee society function score (KSFS) (0–100) *0 being worst, 100 being best***											
Dixon et al. [34]	59	N/A	1 year: 86 5 years: 83	N/A	N/A	N/A	N/A	N/A	N/A	Significant difference in both knee and function scores in the older group (aged ≥ 75 years) (*p* < 0.05). The difference was predominantly due to the functional component of the score.	The results of hydroxyapatite-coated, uncemented TKR in the elderly are clinically comparable with those in a younger group giving a reliable, effective outcome at five years.
Whiteside and Vigano [33]	33	4	1 year: 81 5 years: 83 10 years: 76	1 year: 9 5 years: 8 10 years: 10	N/A	N/A	N/A	N/A	N/A	KSFS significantly improved at 1-, 5-, and 10-year follow-ups.	N/A
Kennedy et al. [24]	Median 45	N/A	Median 1 year: 55 3 years: 55 5 years: 45 10 years: 45	N/A	N/A	N/A	N/A	N/A	N/A	KSFS improved at years 1 and 3, with no difference from the preoperative score at years 5 and 10.	N/A
Severson et al. [30]	Bilateral: 42.47 Unilateral: 43.48	Bilateral: 15.32 Unilateral: 15.42	Bilateral 6 months: 81.57 12 months: 86.07 Unilateral 6 months: 78.80 12 months: 79.73	Bilateral 6 months: 20.66 12 months: 19.54 Unilateral: 6 months: 22.24 12 months: 22.97	N/A	N/A	N/A	N/A	N/A	KSFS significantly improved at 6-month and 1-year follow-up. The improvement in KSFS in the bilateral group was significantly higher compared with the unilateral group at 1-year follow-up.	N/A
Pagnano et al. [35]	26	Range 0–95	33	Range 0–95	N/A	N/A	N/A	N/A	N/A	There was no significant change in the KSFS	N/A
**OKS function (OKSF) (0–28) *0 being worst, 28 being best***											
Townsend et al. [23]	70–79 years: 12.5 >79 years: 11.6	N/A	70–79 years: 16.7 >79 years: 15.4	N/A	70–79 years: 4.2 >79 years: 3.8	N/A	N/A	N/A	N/A	N/A	N/A
Clement et al. [28]	Not reported	N/A	Not reported	N/A	Case (≥80 years): 7.4 Control (65–74 years): 8.0	Case: 5.5 Control: 5.6	3	Distribution-based	Clement et al., 2011 [28]	In both groups, the postoperative OKSF scores at 12 months were significantly improved (*p* < 0.05). Improvement in OKSF scores showed no significant difference between the groups.	The improvement in OKSF scores was significant relative to preoperative scores and was greater than the MCID. The 95% CI for the mean difference in improvement between the groups did not differ by more than the MCID, suggesting there is no clinical difference in outcomes between groups.
**International Knee Score (IKS) Function (0–100) *0 being worst, 100 being best***											
Dowsey et al. [29]	Median 40	IQR 30–50	Median 60	IQR 45–80	Median 25	IQR 10–35	N/A	N/A	N/A	Not reported for significance of pre- and post-operative changes. Only reported difference in change in IKS among groups of non-obese, obese, and morbidly obese.	N/A
**HRS (human retirement study) functional status—Mobility (0–5), Gross motor skills (0–5), Large muscle activities (0–4), Activities of daily living (ADL, 0–4) *0 being best, higher values indicating worse functional status***											
Sloan et al. [31]	Mobility: 1.806 Large muscle: 1.922 Gross motor: 0.816 ADL: 0.384	Mobility: 1.546 Large muscle: 1.196 Gross motor: 1.157 ADL: 0.825	N/A	N/A	Mobility: 0.130 Large muscle: −0.024 Gross motor: 0.012 ADL: −0.084	Mobility: 1.694 Large muscle: 1.256 Gross motor: 1.255 ADL: 0.908	N/A	N/A	N/A	Mobility, gross motor function, and ADL limitations improved among persons receiving TKA relative to the comparison group. The change in large muscle function was not statistically significant.	N/A

**Table 4 jcm-12-01613-t004:** Effect of total knee replacement on general knee outcomes.

Study	Pre-Operative		Post-Operative		Difference		MCID			Summary of Statistical Significance	Summary of Clinical Significance
	Mean	SD	Mean	SD	Mean	SD	MCID (95% CI)	Method	Reference		
**Knee society score (KSS) (0–200), Knee society clinical score (KSCS) (0–100) *0 being worst, 200/100 being best***											
McCalden et al. [25]	KSS: 85	N/A	KSS: 154	N/A	KSS: 69	N/A	N/A	N/A	N/A	There was significant improvement in KSS.	N/A
Dixon et al. [34]	KSS: 94 KSCS: 35	N/A	KSS 1 year: 178 5 years: 174 KSCS 1 year: 91 5 years: 92	N/A	N/A	N/A	N/A	N/A	N/A	Significant difference in knee scores in the older group (aged ≥ 75 years) (*p* < 0.05). The difference was predominantly due to the functional component of the score.	The results of hydroxyapatite-coated, uncemented TKR in the elderly are clinically comparable with those in a younger group giving a reliable, effective outcome at five years.
**Knee society score (KSS) (0–100) *0 being worst, 100 being best***											
Leung et al. [38]	48	16	92	8	46	19	N/A	N/A	N/A	KSS significantly improved at 12 months after TKR; greater improvement observed in the octogenarian group compared with the non-octogenarian group.	N/A
Wendelspiess et al. [37]	54.8	12.5	90.7	8.2	N/A	N/A	N/A	N/A	N/A	KSS significantly improved at 12 months follow-up.	N/A
Whiteside and Vigano [33]	27	10	1 year: 91 5 years: 93 10 years: 92	1 year: 6 5 years: 5 10 years: 8	N/A	N/A	N/A	N/A	N/A	KSS significantly improved at 1-, 5-, and 10-year follow-ups.	N/A
Kennedy et al. [24]	Median 29	N/A	Median 1 year: 91 3 years: 92 5 years: 91 10 years: 90	N/A	N/A	N/A	N/A	N/A	N/A	KSS improved significantly at years 1, 3, 5 and 10.	N/A
Severson et al. [30]	Bilateral: 38.66 Unilateral: 40.51	Bilateral: 12.96 Unilateral: 15.76	Bilateral 6 months: 91.28 12 months: 93.01 Unilateral 6 months: 89.49 12 months: 90.34	Bilateral 6 months: 11.34 12 months: 8.73 Unilateral 6 months: 9.38 12 months: 9.59	N/A	N/A	N/A	N/A	N/A	KSS significantly improved at 6-month and 1-year follow-up. The improvement in the bilateral group was significantly higher compared with the unilateral group at 1-year follow-up.	N/A
**Modified Knee Society Scoring system (0–100) *0 being worst, 100 being best***											
Tankersley and Hungerford [36]	33	Range 0–69	84	Range 28–95	N/A	N/A	N/A	N/A	N/A	There was significant improvement in KSS.	N/A
**Oxford Knee Score (OKS) (0–48) *0 being worst, 48 being best***											
Jiang et al. [22]	70–80 years: 17.5 80+ years: 17	N/A	70–80 years: 1 year: 34.5 5 years: 32.5 10 years: 29.5 80+ years: 1 year: 32.5 5 years: 31 10 years: 28.5	N/A	N/A	N/A	N/A	N/A	N/A	Overall OKS improved significantly at 1, 5, and 10 years after TKR.	N/A
Williams et al. [27]	65–74 years: 20.2 75–84 years: 20.5 ≥85 years: 18.4	65–74 years: 8.0 75–84 years: 8.2 ≥85 years: 8.3	65–74 years 6 mths: 34.5 2 years: 36.7 75–84 years 6 mths: 34.5 2 years: 35.8 ≥85 years 6 mths: 32.0 2 years: 33.3	65–74 years 6 mths: 9.3 2 years: 9.7 75–84 years 6 mths: 8.9 2 years: 9.2 ≥85 years 6 mths: 10.1 2 years: 10.7	65–74 years 6 mths: 14.1 2 years: 16.3 75–84 years 6 mths: 13.7 2 years: 14.8 ≥85 years 6 mths: 13.2 2 years: 14.1	65–74 years 6 mths: 9.8 2 years: 10.4 75–84 years 6 mths: 9.4 2 years: 9.7 ≥85 years 6 mths: 10.4 2 years: 10.3	5	Distribution-based	Murray DW et al. The use of the Oxford hip and knee scores. *J. Bone Joint Surg. Br.* 2007, *89*, 1010–1014 [39]	Postoperative change in OKS showed a linear trend for greater improvement with younger age at both six months (*p* = 0.026) and two years (*p* = 0.014).	The improvement in OKS was greater than the MCID. A clinically significant improvement in OKS was observed in 82.6% of those aged 65–74 years, 82.9% of those aged 75–84 years, and 76.7% of those aged ≥ 85 years at 6 months, and 88.4% of those aged 65–74 years, 84.9% of those aged 75–84 years, and 78.8% of those aged ≥ 85 years at 2 years after TKR. At two years post-operatively the proportion of patients achieving a clinically meaningful improvement was greater in the younger groups (*p* = 0.009).
William et al. [26]	70–80 years: 19.5 >80 years: 18	N/A	70–80 years 1 year: 34 2 years: 34 3 years: 34 4 years: 33.5 5 years: 33.5 6 years: 33 7 years: 33 8 years: 33 9 years: 32 >80 years 1 years: 33 2 years: 33 3 years: 33.5 4 years: 33 5 years: 32.5 6 years: 32.5 7 years: 36 8 years: 34 9 years: 34	N/A	N/A	N/A	N/A	N/A	N/A	Postoperative OKS were significantly higher than pre-operative scores at all time points (*p* < 0.001)	N/A
Clement et al. [28]	Not reported	N/A	Not reported	N/A	Case (≥80 years) 6 mths: 14.0 12 mths: 14.7 Control (65–74 years) 6 mths: 14.2 12 mths: 15.8	N/A	5	Distribution-based	Clement et al., 2011 [28]	In both groups, postoperative OKS scores at 6 and 12 months were significantly improved (*p* < 0.05). Improvement in OKS scores showed no significant difference between the groups.	The improvement in OKS scores was significant relative to preoperative scores and was greater than the MCID. The 95% CI for the mean difference in improvement between the groups did not differ by more than the MCID, suggesting there is no clinical difference in outcomes between groups.
Townsend et al. [23]	70–79 years: 20.3 >79 years: 18.2	N/A	70–79 years: 27.3 >79 years: 26.5	N/A	70–79 years: 7.0 >79 years: 8.3	N/A	N/A	N/A	N/A	N/A	N/A
**International Knee Score (IKS) (0–200), Knee (IKSK 0–100) *0 being worst, 200/100 being best***											
Dowsey et al. [29]	Median IKS:72 IKSK: 32	IQR IKS: 58–89 IKSK: 25–41	Median IKS: 142 IKSK: 86	IQR IKS: 122–173 IKSK: 67–92	Median IKS: 71 IKSK: 49	IQR IKS: 47–92 IKSK: 32–59	N/A	N/A	N/A	Not reported for significance of pre- and post-operative changes. Only reported difference in change in IKS among groups of non-obese, obese, and morbidly obese.	N/A
**KOOS domains: Pain (0–100), Symptoms (0–100), activities of daily living (ADL) (0–100), Quality of life (QOL0 (0–100), Joint replacement (JR) (0–100) *0 being worst, 100 being best***											
Wendelspiess et al. [37]	Pain: 44.8 Symptoms: 48.8 ADL: 51.0 Sports: 21.9 QOL: 25.7	Pain:14.1 Symptoms: 16.3 ADL: 14.8 Sports: 18.4 QOL: 13.1	Pain: 83.7 Symptoms: 80.1 ADL: 85.5 Sports: 65.5 QOL: 72.5	Pain: 15.8 Symptoms: 15.0 ADL: 13.9 Sports: 25.7 QOL: 21.1	N/A	N/A	N/A	N/A	N/A	KOOS scores significantly improved 12 months after TKR. KOOS symptoms and QOL were lower in patients aged <65 years than those aged ≥ 65 years.	N/A
Lyman et al. [21]	Pain: 51 Symptoms: 55 ADL: 55 QOL: 28 KOOS, JR: 53	Pain: 17 Symptoms: 18 ADL: 17 QOL: 17 KOOS, JR: 13	Pain: 88 Symptoms: 83 ADL: 85 QOL: 72 KOOS, JR: 80	Pain: 15 Symptoms: 15 ADL: 16 QOL: 24 KOOS, JR: 14	Pain: 37 Symptoms: 28 ADL: 30 QOL: 44 KOOS, JR: 28	Pain: 20 Symptoms: 22 ADL: 19 QOL: 26 KOOS, JR: 17	Distribution-based: Pain: 8 Symptoms: 9 ADL: 9 QOL: 8 JR: 6 Anchor-based: Pain: 18 Symptoms: 7 ADL: 16 QOL: 17 JR: 14	Distribution-based and anchor-based receiver operating characteristic (ROC) curve methods	Berliner, J.L. *Clin.* *Orthop Relat Res.* 2017, *475*, 149–157 [40] Glassman SD, *J Bone Joint Surg Am.* 2008, *90*, 1839–1847 [41] Paulsen A, Acta Orthop. 2014; *85*: 39–48 [42]	KOOS scores improved significantly 2 years after TKR.	Across KOOS domains and KOOS, JR, the percentage of patients undergoing TKR who achieved an MCID ranged from 77% to 91% using the distribution-based method, and ranged from 76% to 81% using the anchor-based method.
**WOMAC (0–96) *0 being worst, 96 being best***											
Townsend et al. [23]	70–79 years: 46.6 >79 years: 49.2	N/A	70–79 years: 63.5 >79 years: 63.5	N/A	70–79 years: 16.9 >79 years: 14.3	N/A	N/A	N/A	N/A	N/A	N/A
**WOMAC (0–100) *0 being worst, 100 being best***											
Leung et al. [38]	48	20	78	14	29	22	N/A	N/A	N/A	WOMAC improved significantly at 12 months after TKR; similar improvements in octogenarian and non-octogenarian groups.	N/A
McCalden et al. [25]	48	N/A	74	N/A	26	N/A	N/A	N/A	N/A	There was significant improvement in WOMAC.	N/A

**Table 5 jcm-12-01613-t005:** Effect of total knee replacement on health-related quality of life.

Study	Pre-Operative		Post-Operative		Difference		MCID			Summary of Statistical Significance	Summary of Clinical Significance
	Mean	SD	Mean	SD	Mean	SD	MCID (95% CI)	Method	Reference		
**EuroQol (EQ-5D) (0–1) *0 being worst, 1 being best***											
Wendelspiess et al. [37]	0.622	0.169	0.869	0.137	N/A	N/A	N/A	N/A	N/A	There was significant improvement in quality of life.	N/A
Williams et al. [27]	65–74 years: 0.43 75–84 years: 0.45 ≥85 years: 0.42	65–74 years: 0.32 75–84 years: 0.31 ≥85 years: 0.31	65–74 years 6 mths: 0.74 2 years: 0.76 75–84 years 6 mths: 0.74 2 years: 0.74 >85 years 6 mths: 0.70 2 years: 0.66	65–74 years 6 mths: 0.24 2 years: 0.27 75–84 years 6 mths: 0.22 2 years: 0.24 >85 years 6 mths: 0.26 2 years: 0.27	65–74 years 6 mths: 0.31 2 years: 0.32 75–84 years 6 mths: 0.28 2 years: 0.29 >85 years 6 mths: 0.26 2 years: 0.25	65–74 years 6 mths: 0.33 2 years: 0.35 75–84 years 6 mths: 0.32 2 years: 0.33 >85 years 6 mths: 0.33 2 years: 0.32	0.075	Anchor Distribution	Walters SJ et al. Comparison of the minimally important difference for two health state utility measures: EQ-5D and SF-6D. *Qual Life Res* 2005; 14: 1523–1532 [43]	Postoperative change in scores showed a linear trend for greater improvement in the younger age groups at both six months (*p* = 0.012) and two years (*p* = 0.033).	The improvement in EQ-5D was greater than the MCID. A clinically significant improvement in EQ-5D was observed in 72.3% of those aged 65–74 years, 68.7% of those aged 75–84 years, and 68.1% of those aged ≥ 85 years at 6 months, and 73.2% of those aged 65–74 years, 70.1% of those aged 75–84 years, and 67.3% of those aged ≥ 85 years at 2 years after TKR. Younger patients were more likely to have achieved a clinically meaningful improvement at two years (*p* = 0.031).
**EuroQol VAS (0–100) *0 being worst, 100 being best***											
Wendelspiess et al. [37]	62.2	20.5	79.4	14.9	N/A	N/A	N/A	N/A	N/A	There was significant improvement in quality of life.	N/A
Williams et al. [27]	65–74 years: 67.5 75–84 years: 68.3 ≥85 years: 67.6	65–74 years: 22.2 75–84 years: 18.4 ≥85 years: 17.8	65–74 years 6 mths: 77.4 2 years: 75.6 75–84 years 6 mths: 76.0 2 years: 73.4 >85 years 6 mths: 73.9 2 years: 69.9	65–74 years 6 mths: 17.3 2 years: 18.9 75–84 years 6 mths: 16.8 2 years: 18.5 >85 years 6 mths: 20.6 2 years: 18.0	N/A	N/A	N/A	N/A	N/A	Postoperative change in VAS was not significantly different between groups.	N/A
**SF-12 Physical Component Summary (PCS) score (0–100) *0 being worst, 100 being best***											
McCalden et al. [25]	31	N/A	37	N/A	6	N/A	N/A	N/A	N/A	There was significant improvement in PCS score.	N/A
Clement et al. [28]	N/A	N/A	N/A	N/A	Case (≥80 years): 7.9 Control (65–74 years): 10.6	N/A	5.2	Distribution-based	Clement et al., 2011 [28]	Both groups had significant improvement in SF12 PCS score (*p* < 0.05). The control group had significantly greater improvement in comparison to the case group.	The mean difference in improvement of SF12 PCS between the groups did not exceed the MCID, suggesting there is no clinical difference in outcomes between groups.
Dowsey et al. [29]	Median 25.41	IQR 22.42–29.82	Median 35.84	IQR 29.41–45.31	Median 9.47	IQR 3.24–18.84	N/A	N/A	N/A	Not reported for significance of pre- and post-operative changes. Only reported difference in change in SF-12 among groups of non-obese, obese, and morbidly obese.	N/A
Franklin et al. [32]	66–80 years: 30 >80 years: 28	N/A	66–80 years: 55 >80 years: 52	N/A	N/A	N/A	N/A	N/A	N/A	There was significant improvement in PCS score.	N/A
**SF-12 Mental Component Summary (MCS) score (0–100) *0 being worst, 100 being best***											
Clement et al. [28]	N/A	N/A	N/A	N/A	Case (≥80 years): 0.6 Control (65–74 years): 0.4	N/A	N/A	N/A	N/A	SF-12 MCS score improved in both groups but did not reach statistical significance (*p* > 0.2).	N/A
Dowsey et al. [29]	Median 53.51	IQR 43.56–58.51	Median 52.80	IQR 42.20–59.87	Median 0.65	IQR −7.81 to 7.47	N/A	N/A	N/A	Not reported for significance of pre- and post-operative changes. Only reported difference in change in SF-12 among groups of non-obese, obese, and morbidly obese.	N/A

**Table 6 jcm-12-01613-t006:** Satisfaction with the results of total knee replacement.

Study	Satisfaction	Domain	Instrument	Criteria	Follow up (Range)
Wendelspiess et al., 2022. Switzerland [37]	Satisfaction was reported in 89% of patients	Overall satisfaction	5-point Likert scale (very satisfied, satisfied, neutral, unsatisfied or very unsatisfied)	5 points (very satisfied, satisfied, neutral, unsatisfied, or very unsatisfied)	12 months
Clement et al., 2011. United Kingdom [28]	Overall results: 17.4 Case: 19.0, SD 8.1 Control: 18.2, SD 8.1	Overall satisfaction Pain relief Increased function Work/sports Expectations Would you have the operation again? Would you recommend your operation? Hospital experience	6 Point Visual Analogue Scale (1 most satisfied to 6 least satisfied) of 8 questions	Total score ranging from 8 (most satisfied) to 48 (least satisfied)	7.1 (5–10) years
Williams et al., 2013. United Kingdom [27]	Mean (SD) 65–74 yrs: 84.9 (20.8) 75–84 yrs: 85.2 (20.1) ≥85 yrs: 83.6 (19.7) 65–74 yrs: 89.4 (16.2) 75–84 yrs: 87.9 (19.1) ≥85 yrs: 85.9 (16.6)	Overall satisfaction with the surgical outcome Overall satisfaction with the service provision	100 Point Visual Analogue Scale (0 least satisfied to 100 most satisfied)	0 to 100 (least satisfied to most satisfied)	6 months
Pagnano et al., 2004. United States [35]	All but 1 patient was satisfied with the result of TKR	Satisfied with result	Satisfaction assessed but instrument not specified	N/A	3.9 years (minimum 2 years)

## Data Availability

This is a systematic review. No new data were created or analysed in this study. Data sharing is not applicable to this article. All results were presented in the article.

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
