# Peer review of "Patient-Reported Outcomes following Total Knee Replacement in Patients Aged 65 Years and Over—A Systematic Review"

_jcm, 2023, doi:10.3390/jcm12041613_

Round 1
Reviewer 1 Report
The article "Patient-reported outcomes following total knee replacement in patients aged 65 years and over - a systematic review" is devoted to the systematic review of patient-reported outcome assessments after TKR in individuals >65 years of age. The Ovid MEDLINE, EMBASE, and Cochrane libraries were systematically searched to identify studies that examined disease- or health-specific quality of life outcomes after TKR. A wide range of keywords were searched, and the filters used for selection were specified in detail, taking into account the requirements of the Prisma medical selection system. A qualitative synthesis of evidence was also carried out.
The state of the art has been duly analyzed. Most of the cited references are relevant. Their number is equal to 38, but the list of references contains less than 40% of the sources that have been published over the past 5 years. And it is known that, as a rule, publications over the past 5 years contain references to almost all previous studies on the topic in question. But in the case of Article Review, this condition can apparently be omitted. It should also be noted that the authors carried out a broad discussion of the study results.
Reviewer 2 Report
General comments:
This systematic review aims to summarize and investigate the patient-reported outcomes, e.g., pain reduction, function improvement, and life quality, following TKR in individuals aged ≥ 65 years. 18 studies with 20826 patients in total were included in this review, and a clear summary of the effects of TKR on pain, outcomes, life quality, clinical meaning, and satisfaction was drawn respectively. In summary, this systematic review has a careful study design, literature selection, and assessments, thus, this paper is generally persuasive and it can provide valuable information for surgeons to determine a TKR for older patients.
Specific comments:
1. Regarding 1. Introduction: Could you please explain more about the specialties of 65 years old in the context of TKR, and the reason why this paper and the prior study determined 65 years old as a gap, rather than other ages?
2. Complication is one of the most concerning issues for older patients, especially for those TKR patients. I would suggest the authors extract and list the main underlying diseases and complications of patients from the included studies, which may provide hints to the high bias risk of patient-reported outcomes.
3. Although the positive effects of TKR for reducing pain, improving function, and life quality have been confirmed only 1 low bias risk study may lead to an inaccurate conclusion. As the latest included literature is published in September 2022, I would suggest the authors include one more recent study, for instance:
Priol R, et al. Trajectory of chronic and neuropathic pain, anxiety and depressive symptoms and pain catastrophizing after total knee replacement. Results of a prospective, single-center study at a mean follow-up of 7.5 years. Orthop Traumatol Surg Res. 2023 Jan 3:103543. doi: 10.1016/j.otsr.2022.103543. Epub ahead of print. PMID: 36608901.
4. please cite the following reference to discuss how the mental condition would affect the result of the TKA.
Journal of Orthopaedic Surgery and Research 14 (1), 1-8ï¼›Journal of orthopaedic surgery and research 14 (1), 1-10;Journal of Orthopaedic Surgery and Research 14 (1), 1-9
5. The blood loss may affect the result of the TKA. Please cite the following references to discuss.
The Journal of arthroplasty 33 (3), 786-793; JBJS 101 (5), 438-445
Reviewer 3 Report
I thank the authors for conducting this interesting systematic review on the use of PROMS in subjects over 65 undergoing knee arthroplasty.
I have a few observations to make:
- Introduction: in my opinion it should be expanded by discussing what the real challenge is after knee arthroplasty, i.e. rehabilitation. In fact, I noticed that this term does not appear in the entire systematic review, I find that it is a topic that should also be addressed in the light of the Proms: in fact, the patient's outcomes are achieved if at the basis of the resolution of the problem there is not only the choice to resort to surgery but also the awareness that after surgery you will have to face a more or less long period of intensive rehabilitation. At the basis of the rehabilitation process there is the rehabilitation project and the therapeutic alliance with the patient for an action that is centered on the patient's expectations and characteristics.
Materials and methods: from line 102 to line 108 I think he forgot to delete the descriptive guide offered by clinical medicine to direct the action of the authors. Please correct this part.
Results, discussion, and conclusions: the results are presented in a clear way and the discussion and conclusions are coherent but also in this case I ask you why it has not been included whether or not the patients have undergone rehabilitation, whether they have performed it in a specialist center either at home or in a hospital setting. On line 402-403, talk about how COVID-19 has increased waiting lists for knee arthroplasty surgery and has also conditioned the way of doing rehabilitation: in fact, during the pandemic period, many rehabilitation facilities have reinterpreted the way of doing rehabilitation and tele-rehabilitation was used more significantly. The conclusions speak of evidence-based practice: it is based on the evidence present in the literature, the clinician's expertise and above all the patient's beliefs and preferences. A goal is achieved to the extent that it is shared with the patient himself. I beg you to address the issue of rehabilitation at least, even if it is not one of your primary objectives, because in my opinion it is this which, together with the surgery, determines and conditions the outcome of the patient undergoing knee replacement surgery.
I enclose some articles whose reading can facilitate the integration of the rehabilitation theme:
I enclose some articles whose reading can facilitate the integration of the rehabilitation theme:
Timing of rehabilitation on length of stay and cost in patients with hip or knee joint arthroplasty: A systematic review with meta-analysis - PubMed (nih.gov)
Consensus statement for perioperative care in total hip replacement and total knee replacement surgery: Enhanced Recovery After Surgery (ERAS®) Society recommendations - PubMed (nih.gov)
Supervised versus unsupervised rehabilitation following total knee arthroplasty: A systematic review and meta-analysis - PubMed (nih.gov)
Rehabilitation for Total Knee Arthroplasty: A Systematic Review - PubMed (nih.gov)
